# Application of Genomic Selection in Beef Cattle Disease Prevention

**DOI:** 10.3390/ani15020277

**Published:** 2025-01-20

**Authors:** Ramanathan Kasimanickam, Joao Carlos Pinheiro Ferreira, John Kastelic, Vanmathy Kasimanickam

**Affiliations:** 1Department of Veterinary Clinical Sciences, College of Veterinary Medicine, Washington State University, Pullman, WA 99164-6610, USA; joao.cp.ferreira@unesp.br (J.C.P.F.); vkasiman@wsu.edu (V.K.); 2Department of Veterinary Surgery and Animal Reproduction, School of Veterinary Medicine and Animal Science, São Paulo State University, Botucatu 18618-681, Brazil; 3Department of Production Animal Health, Faculty of Veterinary Medicine, University of Calgary, Calgary, AB T2N 1N4, Canada; jpkastel@ucalgary.ca

**Keywords:** beef cattle, breeding programs, disease prevention, genetic resistance, genomics, herd health, precision medicine

## Abstract

Genomic applications in beef cattle disease prevention are revolutionizing how cattle health is managed, offering tools to enhance resistance to common infectious diseases. Through techniques like whole-genome sequencing and genotyping, researchers can identify genetic markers associated with disease resistance, such as to bovine respiratory disease and Johne’s disease. These insights enable more evidence-based breeding programs that select for cattle with superior immune function, reducing reliance on antibiotics and improving animal welfare. Genomic data also support early disease detection and tailored health management strategies, such as customized vaccination programs. Genomic tools can be used to proactively manage herd health, minimize disease outbreaks, and improve biosecurity. Despite challenges of cost and data integration, the potential of genomics to improve disease resistance, reduce economic losses, and enhance sustainability in beef cattle production is substantial. As technologies advance, genomic applications are expected to have increasingly vital roles in livestock management.

## 1. Introduction

The global population is projected to exceed 9.8 billion by 2050 [1,2,3], with rising affluence in developing countries expected to drive increased demand for meat and dairy products [4,5]. However, as global population and demand grow, so too do concerns about the ability of current food production systems to meet these needs, especially given the strain on land and water resources [6,7]. These resources are already scarce, and issues such as biosecurity, particularly infectious diseases, pose further challenges to the sustainability of agricultural systems. Additionally, agricultural production contributes significantly to the emission of greenhouse gases, such as methane, nitrous oxide, and carbon dioxide, exacerbating climate change [8,9,10,11]. To address these mounting challenges, food production systems must improve efficiency in land and water use, reduce greenhouse gas emissions, and enhance resilience to climate variability [8,9,10,11]. One widely suggested solution is the enhancement of the “sustainability” of agricultural practices, which can help balance increased demand with environmental and economic sustainability [6,12,13,14].

The beef cattle industry has a vital role in global food production by providing high-quality protein and other essential products. In the United States, beef production is a critical socio-economic sector [15]. Not only is the U.S. the largest consumer of beef globally, but it also boasts the largest fed cattle industry [16]. The beef production system is structured across three primary sectors: the cow-calf, stocker/backgrounder, and feedlot operations. Cow-calf operations breed cattle and sell weaned calves to stocker operations, which in turn raise them on pasture until they are sold as yearlings. Backgrounding operations feed weaned calves forage-based rations before their entry into feedlots, where cattle are fattened and later sold for harvest.

Despite its importance, the beef industry has numerous challenges, with one of the most pressing being impacts of diseases on cattle health, productivity, and overall welfare. Poor animal welfare due to disease is linked to lower productivity, reduced longevity, poor meat quality, decreased reproductive performance, and higher disease incidence [17]. Disease outbreaks not only cause direct economic losses but also have wider implications for animal welfare, public health, and environmental sustainability [18,19,20,21]. To combat these issues, genomic selection can have a crucial role. By enhancing livestock social behavior, disease resistance, resilience to stress, and adaptability to changes in production systems, genomic technologies offer a promising avenue for improving cattle health and productivity [22].

Biosecurity is an essential element of animal production systems, encompassing a range of strategies to protect animal health, public health, and the environment [23]. It can be allocated into five main compartments: (i) bio-exclusion, which prevents pathogens from entering a farm; (ii) bio-compartmentalization, which limits pathogen spread within a farm; (iii) bio-containment, which prevents transmission to other farms; (iv) bio-prevention, which focuses on controlling zoonotic diseases that can spread to humans; and (v) bio-preservation, which safeguards against environmental contamination. Genomic applications have key roles in all these compartments [23]. For example, quarantining newly introduced animals aids in bio-exclusion, whereas genomic applications for disease prevention in cattle address all five compartments.

Disease resistance is a critical factor influencing cattle health, welfare, and productivity. It is determined by a combination of genetic, immune, environmental, and management factors [24]. Genetic resistance to disease, in particular, can be enhanced through genomic selection. By identifying genetic markers associated with disease resistance or susceptibility, genomic tools can identify cattle with better disease resistance, improving herd health and reducing reliance on antibiotics and other interventions [25,26,27,28]. Combining genomic tools with immunological, clinical, and environmental data provides a more comprehensive approach to assessing disease resistance and developing effective management strategies.

This review explores the role of genomics in disease prevention in beef cattle. It examines how genomic applications can identify genetic markers linked to resistance or susceptibility to specific diseases, improve breeding programs, and enhance herd management. Integration of genomic data into beef cattle management is not only a promising strategy for improving cattle health but also a crucial step toward achieving more sustainable, resilient, and productive beef production systems. By leveraging genomic technologies, the beef industry can move closer to meeting growing global demands for meat while minimizing environmental impacts and enhancing animal welfare.

## 2. The Role of Genomics in Preventing Diseases in Beef Cattle

Genomics, the study of an organism’s complete genetic makeup, provides insights into the genetic factors that influence an animal’s susceptibility or resistance to diseases [29]. In cattle, genomic tools enable breeders and veterinarians to understand the genetic basis of disease resistance, identify biomarkers linked to diseases, and implement genomic selection to enhance disease resilience in herds [30]. These tools can be used to target complex diseases caused by pathogens, environmental factors, or genetic predispositions.

Genomic applications also enable a more efficient approach to disease prevention by identifying animals with genetic traits that confer natural resistance to certain diseases, reducing the need for chemical interventions like antibiotics and vaccines [31]. By focusing on the genetic components of disease resistance, genomic tools can enhance both herd health and farm profitability [32].

Various methods have been used to control diseases, including vaccinations, medical treatments, and eradication strategies, all of which have had varying success. However, these approaches face several challenges, including side effects/failure of vaccines, concerns about drug residues and resistance associated with medical treatments, increased culling, morbidity, decreased production, increased mortality, and high costs [33,34,35,36,37]. In addition to these methods, emerging technologies like genome editing, biosensors, and probiotics offer new alternatives for improving animal health [38,39,40,41,42]. Genomic selection is revolutionizing livestock breeding, offering substantial advancements in disease resistance and food safety. Genomic tools to identify beneficial genetic traits can create healthier animals that are less prone to disease, require fewer pharmaceuticals, and produce safer food. These innovations improve animal well-being and also increase the sustainability and safety of the global food supply [43], which are essential for meeting growing demands for food in the face of increasing global challenges.

## 3. Genomic Tools and Techniques for Disease Prevention

Several genomic technologies and techniques (Figure 1) are particularly useful in understanding and preventing diseases in beef cattle [30,44]. These include genomic selection, genome-wide association studies (GWASs), functional genomics, and next-generation sequencing (NGS). Each of these tools provides critical insights into the genetic underpinnings of disease resistance, enabling more informed management practices [30,44]. However, genomic selection efficiency of beef cattle is lower compared to dairy cattle. This could be attributed to various factors such as breed heterogeneity, less sophisticated breeding programs and structures, the prevalence of natural service, cross-breeding in commercial herds, and effective population size [45]; the latter refers to the number of individuals that actively contribute to the next generation within a population [46].

### 3.1. Genomic Selection

Genomic selection uses DNA data to estimate the genetic potential of an individual animal for specific traits, including disease resistance. By analyzing genetic variations, particularly single nucleotide polymorphisms (SNPs), genomic selection can identify animals with favorable genotypes that are more resistant to specific diseases [47,48]. This facilitates early-life selection for disease resistance, even before exposure to pathogens.

For example, cattle with genetic markers linked to resistance to diseases like bovine respiratory disease (BRD) or mastitis can be preferentially bred to improve herd health [30]. Genomic selection facilitates rapid incorporation of these traits into breeding populations, particularly important for diseases with substantial economic impacts [30,49,50]. An animal’s response to complex diseases like BRD is influenced by multiple factors, including the environment, the pathogen, and the animal’s genetics. Heritability of resistance to BRD ranges from 0.07 to 0.29 [51]. Additionally, it has been suggested that an epigenetic component may have a role, as disease episodes early in a calf’s life negatively affect lifetime performance and age at first calving. Consequently, studying methylation profiles of these calves could provide valuable insights for future breeding programs aimed at improving herd resistance, while also enhancing understanding of mechanisms underlying disease susceptibility [52,53].

Genomic selection offers great promise but has several limitations. One major concern is the potential reduction in genetic diversity, which can limit adaptability to changing environments or new diseases. Many traits are polygenic and influenced by environmental factors, making them difficult to predict accurately. Additionally, genomic selection often relies on existing genetic data, which may miss rare variants or fail to capture all factors influencing traits. While effective when large, high-quality datasets are available, these can be costly and time-consuming to obtain. There is also a risk of unintended consequences, such as the selection of traits that inadvertently cause health problems or reduced fertility. Furthermore, genomic models often overlook environmental factors, which can lead to over-reliance on genetic traits that may not perform well in different conditions. Selection bias can also arise if the data used are not representative of the broader population. Finally, despite its power, genomic selection is not foolproof and can struggle with complex traits that involve intricate gene–environment interactions. These challenges emphasize the need for careful implementation and ongoing research.

### 3.2. Genome-Wide Association Studies (GWASs)

GWASs are a powerful tool for identifying genetic loci associated with disease resistance or susceptibility in beef cattle. GWAS involves scanning the genome for SNPs linked to specific diseases or immune responses [54]. By identifying these markers, researchers can pinpoint genes involved in disease resistance, providing valuable targets for genomic selection.

For instance, studies on resistance to diseases like Johne’s disease (a chronic infection caused by *Mycobacterium avium* subspecies *paratuberculosis*) and leptospirosis have identified genetic markers linked to resistance [55,56,57]. These markers can be used to select animals with naturally higher resistance to these diseases, reducing reliance on pharmaceutical treatments. However, GWASs have limitations, including a focus on common variants, which often miss rare variants with potential effects. The identified genetic variants typically have small effect sizes, explaining only a small portion of the genetic variation for traits. GWAS results can also be confounded by population stratification, and they often fail to account for environmental factors or gene–environment interactions. Additionally, many identified variants lack clear biological relevance, requiring further investigation.

### 3.3. Functional Genomics

Functional genomics focuses on understanding the roles of specific genes in disease resistance. This involves studying expression of genes related to immune responses and their role in protecting cattle from pathogens [58]. Transcriptomics enables identification of genes upregulated in response to infections, providing insights into how cattle mount immune responses for specific diseases. For instance, the immune response to Porcine Reproductive and Respiratory Syndrome Virus (PRRSV) varied strongly among individual pigs and was considered for potential genetic selection [59,60].

Immune-related genes, such as those involved in the Toll-like receptor signaling pathway, have crucial roles in defending cattle against bacterial and viral infections [61,62]. Functional genomics can provide deeper understanding of how genetic variation influences disease resistance, ultimately leading to development of more resilient cattle populations [63].

Functional genomics offers valuable insights but also has several limitations. One major challenge is the complexity of gene function, as genes rarely act in isolation and often interact with other genes and environmental factors, making their roles difficult to pinpoint. Additionally, there is an incomplete understanding of non-coding regions of the genome, which are crucial for gene regulation but poorly characterized. Functional genomics also relies on experimental models, such as cell lines, which may not accurately reflect animal biology. Technologies like CRISPR and RNA sequencing are prone to off-target effects and data gaps, complicating result interpretation. Data integration across different platforms (e.g., transcriptomics, proteomics) is resource-intensive and can lead to conflicting conclusions. Moreover, large sample sizes are often required, making studies costly and time-consuming. Finally, ethical concerns about gene manipulation and potential misinterpretation of data further complicate the field. These limitations underscore the need for careful validation and complementary approaches in functional genomics.

### 3.4. Next-Generation Sequencing (NGS)

NGS technologies, which enable high-throughput sequencing of genomes, have revolutionized genomic research in livestock [64]. NGS facilitates identification of rare genetic variants and mutations associated with disease resistance [65]. For example, NGS has been used to sequence genomes of cattle with high resistance to diseases like bovine viral diarrhea (BVD) or foot-and-mouth disease (FMD), identifying key genetic variants that could be used in selective breeding programs [66].

NGS can also help identify genetic variations related to pathogen interactions, such as specific receptors that pathogens use to enter cells. Understanding these interactions will inform development of more effective breeding strategies to enhance disease resistance [30,65].

NGS has transformed genomics but comes with several limitations. The high cost of sequencing and the need for significant computational resources for data analysis can be barriers, particularly for large-scale studies. NGS is also prone to sequencing errors, such as base substitutions or indels, which can complicate data interpretation, especially in complex or repetitive regions. Low-coverage or poor-quality data in certain areas, like guanine and cytosine (GC)-rich or repetitive sequences, can lead to gaps or inaccuracies in genome assemblies. Additionally, the short read lengths of many NGS technologies can make it difficult to assemble genomes accurately, particularly in regions with structural variation. Variant interpretation remains challenging, as the biological significance of many variants, especially in non-coding regions, is unclear. In clinical applications, the lack of standardized protocols can introduce variability in results. Finally, ethical and privacy concerns arise with the handling of large genomic datasets, especially regarding consent and misuse of personal genetic information. These challenges highlight the need for improvements in NGS technology, data analysis, and ethical guidelines.

## 4. Genomic Approaches to Specific Diseases

### 4.1. Bovine Respiratory Disease (BRD)

Bovine respiratory disease (BRD) is a leading cause of morbidity and mortality in beef cattle, particularly in feedlots. It has been reported that 60.6% of slaughtered feedlot cattle never treated for BRD had lung lesions present [67] and 68% of slaughtered feedlot steers with no recorded history of BRD presented with lung lesions [68]. BRD is often caused by a combination of bacterial, viral, and environmental factors. Genetic susceptibility of cattle to BRD varies widely, and genomic tools can identify animals that are naturally more resistant to the disease.

Through genomic selection and GWAS, researchers have identified SNPs associated with increased resistance to BRD, enabling selection of cattle that are less prone to developing the disease [69,70,71]. Additionally, functional genomics research has shed light on immune system pathways that protect against BRD, such as the role of the major histocompatibility complex (MHC) in pathogen recognition [72,73,74].

### 4.2. Mastitis

Mastitis, an inflammation of the mammary gland often caused by bacterial infections, is another important health issue in beef cattle, especially in dairy-beef crossbreeds [75,76,77]. The disease reduces milk production and quality, affecting both animal welfare and farm profitability. Adjusted 205 d weight gain for calves with *Staphylococcus aureus*-infected dams was 9.6 kg less than for calves with uninfected dams [78].

Genetic selection for mastitis resistance is a powerful tool in improving cattle health and farm profitability. Leveraging genomic data to identify and breed animals with superior disease resistance can reduce mastitis incidence, improve milk quality, and decrease reliance on antibiotics. Genetic resistance to mastitis is heritable, and genomic tools allow breeders to select animals based on genetic markers associated with immune function and udder health, such as somatic cell count (SCC) and electrical conductivity (EC) [79,80,81].

Somatic cell count (SCC), a measure of the number of white blood cells in milk, is a key indicator of subclinical mastitis. Somatic cell score (SCS) is a logarithmic conversion of the SCC, essentially placing the cell count on a more manageable linear scale where each increase in score represents a doubling of the cell count. Monardes et al. (1983) reported that heritability of SCC and SCS were ~6 and 12%, respectively [82]. Thus, higher heritability is another major advantage for using SCS for computing genetic evaluations. In other recent studies using DHI data, heritability of SCS was ~10 to 12% [83,84].

Electrical conductivity in milk increases due to altered concentrations of anions and cations (K^+^, Na^+^, Cl^−^) after bacterial infection of the udder [79] and is used for indirect selection against mastitis [80]. Heritability of EC ranged from 0.02 to 0.11.

Sufficient evidence exists that genetic factors underlie resistance or susceptibility to mastitis. Therefore, selection based on breeding values for quantitative traits, e.g., SCC or EC, or selection based on candidate genes, e.g., quantitative trait locus or MHC, in conjunction with non-genetic methods of herd health management, e.g., vaccination, will reduce the frequency of mastitis. Genomic studies have identified several genetic loci related to the immune response in the udder, which influence susceptibility to mastitis in dairy cattle and could also be applied to beef cattle [85,86].

Genotyping of cows identified 1013 genes associated with udder morphology, mastitis, and production phenotypes (e.g., *ESR1*, *FGF2*, *FGFR2*, *FGFR2*, *GLI2*, *IQGAP3*, *PGR*, *PRLR*, *RREB1*, *BTRC*, and *TGFBR2*) [87]. Conversely, other authors reported that the genes *AKNA*, *MIR455*, *ORM1*, and *GNG10*, located on chromosome 8 where SNP rs41609496 is located, were associated with functions on immunity [88].

A systematic review and gene prioritization analysis of GWAS was performed to identify potential functional candidate genes associated with resistance to mastitis-related traits in dairy cattle [85] and reported that 24 genes (*ABCC9*, *ACHE*, *ADCYAP1*, *ARC*, *BCL2L1*, *CDKN1A*, *EPO*, *GABBR2*, *GDNF*, *GNRHR*, *IKBKE*, *JAG1*, *KCNJ8*, *KCNQ1*, *LIFR*, *MC3R*, *MYOZ3*, *NFKB1*, *OSMR*, *PPP3CA*, *PRLR*, *SHARPIN*, *SLC1A3*, and *TNFRSF25*) were associated with both SCC and clinical mastitis-related traits.

By incorporating these genetic markers into breeding programs, beef producers can select cattle with enhanced resistance to mastitis [89]. This should reduce the need for antibiotics and other treatments, improving both animal health and product quality.

### 4.3. Calf Diseases

Beef operations strive to produce calves that are robust and capable of thriving, critical for both calf well-being and economic sustainability [90,91]. Many studies have quantified genetic parameters for health and vitality traits in cattle, particularly beef calves [92]. Heritability estimates for calf vigor range from 0.01 to 0.09 [93,94], with similar heritability estimates for scours and pneumonia [23,38,92]. Heritability estimates for subjectively scored calf birth weight tend to be greater, ranging from 0.17 to 0.22 [91,95]. Condon et al. (2021) reported heritability estimates for vigor, birth weight, and pneumonia were 0.12 (0.02), 0.33 (0.03), and 0.08 (0.02), respectively, but no genetic variance was detected for scours [91].

Many beef production systems are highly dependent on antimicrobials or anthelmintics to maintain animal health and welfare. However, there is increasing impetus to reduce or avoid these products, without compromising animal welfare.

### 4.4. Tuberculosis and Clostridial Diseases

Gene editing enables precise modification of an animal’s genome, offering substantial potential to improve various aspects of livestock management, particularly in beef cattle [96]. By directly altering specific genes, it is possible to enhance cattle resistance to common diseases, e.g., respiratory infections, digestive disorders, and other pathogen-related conditions [30]. These genetic modifications can also improve overall animal health, reducing disease outbreaks. Moreover, gene editing can improve calf welfare by making them more resilient to environmental stressors, enhancing growth rates, and boosting overall well-being [97]. Long-term benefits of gene editing in cattle include not only healthier animals but also more sustainable farming practices, as reducing the need for antibiotics and other interventions can lower the environmental footprint of beef production. Through these advancements, the agricultural industry can move toward more efficient and ethical practices, ensuring healthier livestock while maintaining productivity.

Disease is a major threat to the livestock industry, with outbreaks often causing substantial economic losses. However, rapid progress in CRISPR/Cas9 technology has introduced a promising solution for addressing livestock diseases [98,99]. This gene-editing technology is an effective way to combat disease transmission, reduce the risk of epidemics, lower treatment expenses, prevent outbreaks, and ensure food safety [97].

Bovine herpesvirus type 1 (BHV-1), a virus responsible for infectious rhinotracheitis, vulvovaginitis, and reproductive losses in cattle, poses a serious threat to livestock health [100]. To improve control over BHV-1, CRISPR/Cas9 was used to knock out the virus’s UL41 gene, a major virulence factor, significantly reducing the virus’s ability to cause harm [101]. Disrupting this gene greatly diminished the virus’ pathogenic effects [102].

Moreover, gene-editing technologies have promise in enhancing the immune systems of cattle, offering potential protection against bovine tuberculosis, a zoonotic disease caused by *Mycobacterium bovis* that poses serious risks to both public health and livestock. In one study, homology-mediated end joining (HMEJ) technology was used to target the bovine ROSA26 locus, successfully inducing expression of the *NRAMPI* gene, and creating genetically edited cattle with improved resistance to tuberculosis [103,104].

The KALRN gene is associated with resistance to bacterial infections in ruminants. In an in vitro study investigating the effects of rs384223075 (RS-075) deletion in *KALRN* intron 5 on occurrence of *Mycobacterium bovis* and *Brucella abortus* infections in cattle, the RS-075 deletion was linked to an enhanced cellular response to bacterial antigens and unspecific stimulation in mononuclear cells derived from beef crossbred cows; this was consistent with other reports that supported an important role of the *KALRN* gene and its polymorphisms in host responses to intracellular pathogens [105]. Collectively, these findings underscored the potential of gene editing to bolster cattle immunity against tuberculosis.

Additionally, *Clostridium chauvoei*, an anaerobic, Gram-positive bacterium that causes blackleg disease in livestock, produces *CctA*, a key hemolysin and cytotoxin crucial to the bacterium’s ability to infect and cause disease [106]. Recent efforts used CRISPR/Cas9 technology to develop mutants of *C. chauvoei* with edits to the *CctA* gene, significantly reducing its cytotoxicity and, in turn, lowering the risk of infection to both livestock and humans [107,108].

### 4.5. Johne’s Disease

Johne’s disease, a chronic, infectious disease of cattle caused by the bacterium *Mycobacterium avium* subspecies *paratuberculosis*, can cause severe economic losses due to decreased productivity and increased culling [109,110,111]. Genomic research has identified genetic markers associated with resistance to Johne’s disease, enabling selection of cattle that are less likely to contract the disease or experience severe symptoms [112,113,114].

It is widely acknowledged that not all asymptomatic animals will develop clinical symptoms of Johne’s disease during their productive lives [108,109,110]. Identifying genetic markers linked to tolerance to paratuberculosis could assist in determining which infected cows should be retained rather than culled, thereby improving the cost-effectiveness of control programs [115]. Furthermore, incorporating new genetic variants associated with paratuberculosis susceptibility and tolerance into marker-assisted breeding programs could help producers select cattle that are both less susceptible to paratuberculosis and more resistant to other bovine diseases [115,116,117]. This approach would ultimately reduce economic losses and lower reliance on antimicrobial treatments. Preventing endemic and chronic diseases like paratuberculosis through selection of resilient cattle is crucial for promoting sustainable and efficient cattle production, while also supporting the rural economy [118,119]. Although this strategy would take time to fully implement, the long-term benefits of breeding resilient animals could be enduring.

### 4.6. Enteric Diseases

Genetic selection for enteric disease and parasite resistance in beef cattle is an increasingly important strategy for improving herd health, productivity, and sustainability in livestock farming [30,120]. Enteric diseases, caused by pathogens such as *Escherichia coli*, Salmonella, and Clostridium spp., alongside parasitic infections from organisms like *Haemonchus contortus* and *Ostertagia ostertagi*, can significantly impact cattle performance [121]. These infections can result in reduced weight gain, lower feed efficiency, increased veterinary costs, and in severe cases, animal death. Traditional approaches to managing these diseases often rely on antibiotics and anthelmintics, but concerns about antimicrobial resistance, environmental impacts, and rising treatment costs have stimulated interest in alternative strategies such as genetic selection [122,123,124].

Advances in genomic technologies, including genome-wide association studies (GWASs) and high-throughput sequencing, have identified specific genetic markers linked to resistance against both enteric diseases and parasitic infections [124,125]. These traits may influence immune response, gut health, and the resilience of cattle to internal parasites by enhancing the ability to control worm burdens or reduce pathogen loads in the gastrointestinal tract [126]. Selection for these genetic traits can reduce reliance on chemical treatments and promote sustainability.

Certain cattle breeds and individual animals have better natural resistance to gastrointestinal parasites, with some exhibiting an ability to maintain lower parasite loads even in environments with heavy worm challenges [127,128]. Similarly, cattle with enhanced immune responses are more likely to recover quickly from enteric diseases or resist infections in the first place [129]. Selecting for these traits can not only reduce the incidence of disease but also improve overall resilience and productivity of the herd.

Animal genetic and physiological factors influence the persistence of *E. coli* O157 in cattle at high concentrations [130]. *STM3602* and *STM3846* genes represent an exciting new class of virulence determinants directly linking these genes to colonization of Salmonella in the small intestines of cattle [131]. CRISPR-mediated homology-directed repair and somatic cell nuclear transfer were used to produce a live calf with a six-amino-acid substitution in the BVDV binding domain of bovine *CD46* [132].

Incorporating both enteric disease resistance and parasite resistance into breeding programs offers several benefits: healthier cattle, fewer veterinary interventions, and greater environmental sustainability. Furthermore, by improving the genetic makeup of cattle with respect to these challenges would foster long-term improvements in herd health, ensuring better performance despite environmental stressors such as changing diets or climate conditions. As research advances, the potential for even more precise and effective genetic selection for disease and parasite resistance will continue to grow, offering a promising pathway to more robust and sustainable beef cattle populations.

### 4.7. Brucellosis

Genetic selection for cattle to counter brucellosis is a promising long-term strategy for reducing the disease’s impact. By identifying and breeding animals with innate resistance to Brucella infection, including those with robust immune responses, cattle populations can improve over time [105,133,134]. Certain breeds, such as Brahman cattle and some indigenous African and South American breeds, have higher resistance to brucellosis [133,134]. Advances in genomic technologies enable marker-assisted and genomic selection to pinpoint genes associated with brucellosis resistance, though the trait is complex and likely involves multiple genes. Further research, using standardized in vivo phenotyping protocols, is needed to evaluate *SLC11A1* 3’UTR polymorphisms as a basis for selecting naturally resistant ruminants [135,136,137]. While breeding for resistance is important, it must be combined with other control measures e.g., vaccination, biosecurity, and regular testing to prevent disease spread. Despite challenges such as the complexity of resistance and the cost and time required for genetic improvements, this approach has the potential for more resilient herds and reduced economic losses from brucellosis.

## 5. Integration of Genomic Selection and Assisted Reproductive Technology

Genomic selection in beef cattle faces challenges like limited phenotyping, high diversity of breeds and crossbreds, and underdeveloped breeding programs, particularly due to limited use of artificial insemination and fewer offspring per female [138]. Incomplete recording of economically important traits and a lack of comprehensive international data further hinder progress. Despite these challenges, genomic selection can be a practical and effective alternative to traditional breeding methods in beef cattle [139].

Integration of genomic selection and precision mating using assisted reproductive technology (ART) (Figure 2) is revolutionizing livestock breeding by providing a more efficient and targeted approach to genetic improvement [140]. Artificial insemination (AI), embryo transfer (ET), in vitro fertilization (IVF), and cloning have a complementary role by enabling rapid reproduction of genetically superior animals. ART enables efficient transfer of high-quality genetics across a broader population [22,141] that would otherwise be limited by natural breeding. For example, AI allows widespread dissemination of semen from top-tier sires, whereas ET and IVF enable production of large numbers of genetically superior offspring from select donor females.

Genomic selection involves using genomic data to predict the genetic potential of animals for specific traits. These traits may include disease resistance, growth rate, reproductive efficiency, meat quality, and feed conversion efficiency. Genomic testing will identify animals with superior genetic profiles at a younger age, making it possible to make breeding decisions much earlier in an animal’s life compared to traditional methods.

Genetic selection and ART are a powerful combination that accelerates genetic progress [142]. Genomic selection identifies individuals as donors or sires for ART procedures, optimizing selection [143]. This leads to more informed and precise breeding decisions, targeting animals that are genetically predisposed to excel in desired traits (Figure 3). The synergy between genomic selection and ART can reduce generation intervals, enabling faster genetic gains. Traditionally, achieving genetic improvements in a herd could take several years, but by using GS to identify superior animals early and ART to increase their genetic contribution, these improvements can occur much more quickly [144].

Another advantage of combining genomic selection and ART is the ability to increase selection intensity, i.e., the proportion of genetically superior animals in a breeding population [22,141]. Selecting top-performing individuals based on genomic data and using ART to maximize reproductive output facilitates a rapid increase in herd genetic quality, particularly with limited numbers of breeding animals or when there is a need to address specific traits, such as disease resistance or climate resilience, in a short interval.

Integration of genomic selection and ART also has substantial potential for improving animal health and welfare. For instance, genomic selection can identify animals with resistance to specific diseases, and ART can ensure that these disease-resistant traits are rapidly propagated [22,65]. This approach reduces reliance on chemical treatments, vaccines, and antibiotics, leading to healthier animals and more sustainable production systems. Additionally, ART techniques (e.g., IVF and ET) can address some fertility issues.

On a broader scale, combining GS and ART contributes to more sustainable livestock production systems by reducing the environmental footprint. By improving genetic traits such as feed efficiency, disease resistance, and reproductive success, these technologies can reduce the resources required, such as land, water, and feed. Moreover, faster genetic improvements can help livestock adapt to changing environmental conditions, such as climate variability, without the need for additional resources.

## 6. Challenges and Limitations

Although genomic applications in disease prevention offer substantial benefits, several challenges and limitations remain [145,146]. A major challenge is the cost of genomic testing, particularly for small-scale operations [146]. The cost of genotyping has decreased substantially but may still be prohibitively expensive for some.

Another challenge is the complexity of the genetic architecture of disease resistance. Many diseases in cattle are influenced by multiple genes, making it difficult to identify genetic markers that can be used for reliable prediction [147]. Moreover, environmental factors, management practices, and pathogen variability also have important roles in disease outbreaks, complicating the relationship between genetics and disease resistance. Interactions between diseases are expected [147]; therefore, selection for resistance to one disease may increase susceptibility to other diseases.

Finally, integrating genomic data into breeding programs requires sophisticated bioinformatics tools and expertise, as well as a comprehensive understanding of genetic variation within herds [47]. Properly implementing genomic selection for disease resistance requires collaboration among breeders, veterinarians, and researchers, as well as continuous monitoring of disease prevalence and genetic performance [147].

## 7. Future Directions

The future of genomic applications in beef cattle disease prevention is highly promising, offering a range of exciting possibilities for improving animal health and productivity. Advances in functional genomics, next-generation sequencing (NGS) technologies, and bioinformatics are rapidly advancing our understanding of the genetic basis of disease resistance in cattle. These innovations enable researchers to explore relationships among genes, immune responses, and disease susceptibility, providing deeper insights into how specific genetic traits help cattle combat diseases. In addition, the information may improve management of lethal recessive alleles [148]. Increasing genomic data will inform development of more precise, targeted breeding strategies that focus on disease resistance. These strategies can reduce the need for antibiotics and other pharmaceutical interventions, ultimately leading to healthier herds and more sustainable beef production.

One of the most exciting areas of genomic research is integration of genomic selection with complementary technologies such as precision farming, sensor technologies, and artificial intelligence (AI). By combining genomic data with real-time health monitoring tools, these technologies could revolutionize cattle management [149,150,151,152,153,154,155]. Precision farming techniques, including GPS tracking and automated feed systems, can collect data on environmental factors and individual animal behavior, enabling a more comprehensive understanding of cattle health [152,153,154]. AI and machine learning algorithms can analyze vast amounts of data and identify patterns that may indicate early signs of disease or stress, even before symptoms appear [154,155]. This real-time monitoring system could not only help in early disease detection but also enable implementation of personalized breeding programs tailored to the unique genetic makeup of each animal [156,157]. Precision-based approaches promise to optimize genetic selection, improve disease resistance, and enhance herd productivity and animal welfare [156,157].

Furthermore, development of genomics-based vaccines and therapies could be a game-changer in the fight against cattle diseases. Understanding genetic factors that influence how cattle interact with pathogens and mount immune responses will inform development of vaccines that are more effective and specific to genetic characteristics of various cattle breeds or even individual animals [30]. Genomic insights could also lead to creation of novel therapeutic treatments, such as gene therapies or immune modulators, that enhance natural defense mechanisms of cattle, providing long-term protection against a broad range of diseases. In conjunction with more efficient and widespread disease surveillance systems, these innovations could dramatically improve disease prevention strategies, ensuring that beef cattle remain healthy throughout their lifecycles.

Genomics has the potential to address some of the most persistent and costly diseases in beef cattle. Pinpointing genetic markers associated with resistance to these diseases will enable selection of animals that are naturally less susceptible, reducing the need for antibiotics and minimizing risks of antimicrobial resistance. As genomic technologies become more integrated into herd management practices, producers will have access to powerful tools that enable them to make data-driven decisions, enhancing the sustainability of beef production while safeguarding animal health and welfare.

Integration of genomics into disease prevention strategies also offers a pathway to reducing environmental impacts of beef production. Healthier cattle that are more resilient to disease will require fewer interventions and less intensive care, reducing the overall environmental footprint of the industry [157]. Furthermore, a focus on disease resistance could improve feed utilization and growth rates, and reduce mortality, all of which contribute to more sustainable beef production [18,31,158,159].

Although challenges remain, such as the cost of genomic testing, the need for specialized expertise in data interpretation [160], and complexities of managing genetic diversity within herds [161], the potential of genomic applications in disease prevention is vast. As technologies evolve and become more affordable, and as knowledge of cattle genetics continues to improve, genome-driven disease prevention will have an increasing role in shaping the future of the beef cattle industry [162]. These innovations will not only enhance cattle health and productivity, but also help address the industry’s long-term sustainability goals, ensuring that beef production can meet growing global demands, promote animal welfare, and minimize environmental impact [163].

## 8. Conclusions

Genomic applications in beef cattle disease prevention are revolutionizing the way the industry approaches herd health and disease management. Integrating advanced tools such as genomic selection, genome-wide association studies (GWASs), functional genomics, and next-generation sequencing (NGS) technologies facilitates identification of individual animals with greater resistance to specific diseases. This proactive approach reduces reliance on antibiotics, vaccines, and other interventions that are commonly used in traditional disease management practices, leading to a more sustainable and cost-effective system. Moreover, these genomic tools enable targeting traits associated with disease resistance at a genetic level, offering the possibility of breeding cattle that are inherently less susceptible to common illnesses like bovine respiratory disease, mastitis, or foot rot.

As scientific knowledge about the genetic basis of disease resistance continues to grow, genomic applications will become increasingly central to efforts aimed at improving cattle health, overall productivity, and animal welfare. This deeper understanding of how certain genes influence immune function and disease resistance opens new avenues for creating herds that can thrive in diverse environmental conditions and production systems, while simultaneously reducing the need for chemical interventions that can have long-term ecological and health implications.

Despite the promising potential, several challenges remain in widespread adoption of genome-driven disease prevention strategies. A major hurdle is the high cost of genomic testing, which can be prohibitive, especially for smaller operations. Additionally, interpreting complex genetic data and translating it into actionable breeding decisions requires expertise and resources. There are also concerns regarding the long-term sustainability of genetic selection, especially if it inadvertently decreases genetic diversity.

Nevertheless, the potential for genome-driven disease prevention in beef cattle is vast, offering innovative solutions to some of the most persistent health challenges facing the industry. As research progresses, and as genomic technologies become more accessible and affordable, the beef cattle industry should benefit from healthier, more resilient animals, ultimately leading to improved productivity, reduced environmental impact, and enhanced animal welfare. In the long term, integration of genomics into disease management could reshape the future of beef production, making it more sustainable, efficient, and humane.

## Figures and Tables

**Figure 1 animals-15-00277-f001:**
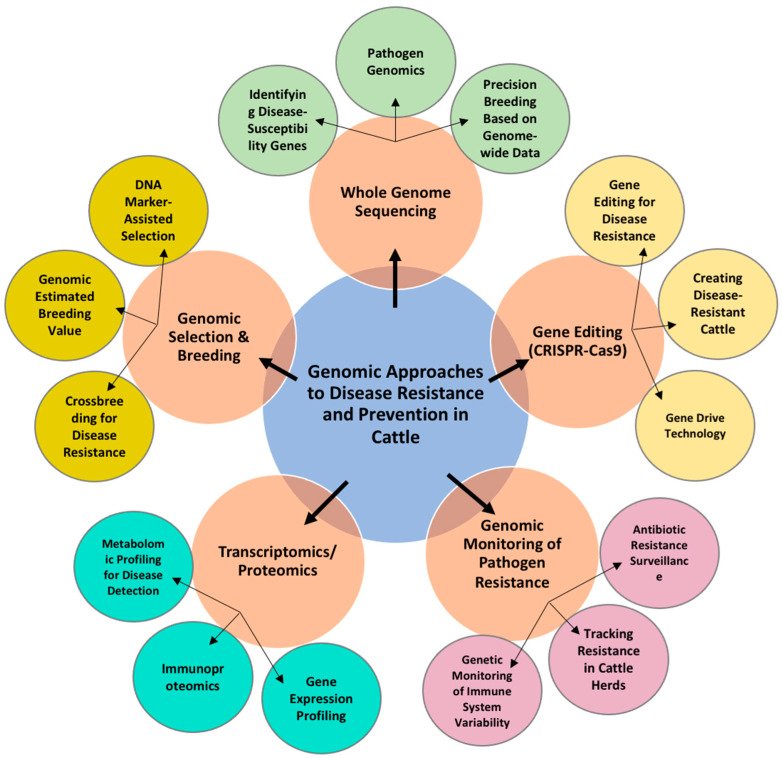
A comprehensive illustration of various genomic tools and techniques that interact to prevent disease in cattle, enhancing both scientific understanding and practical applications in the livestock industry.

**Figure 2 animals-15-00277-f002:**
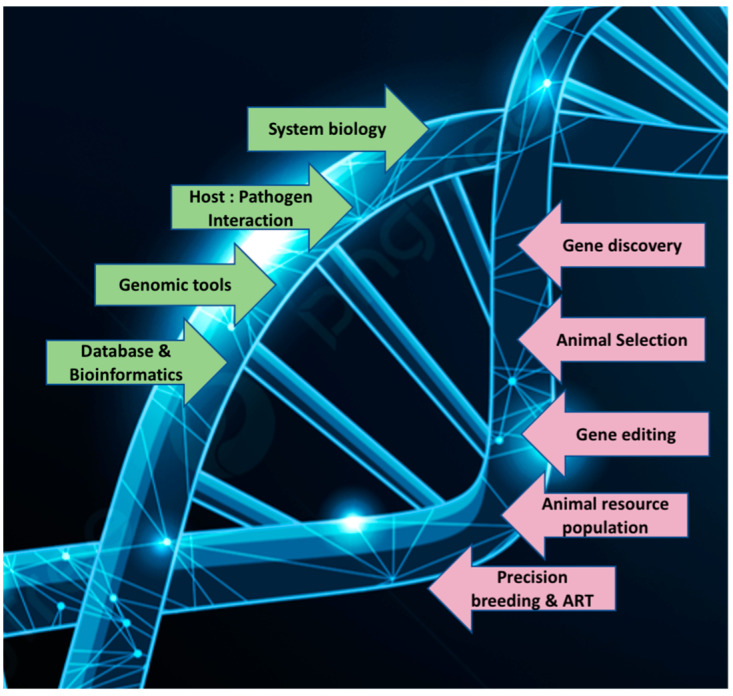
Illustration of integration of genomic data into large-scale genetic evaluation programs and the use of genomic information to design precision mating systems.

**Figure 3 animals-15-00277-f003:**
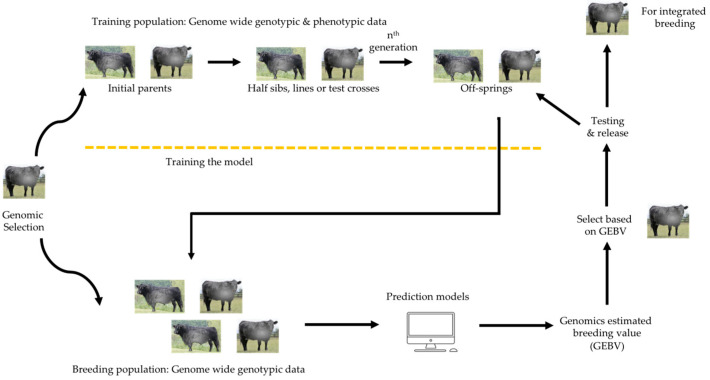
Schematic representation of steps involved in genomic selection and breeding.

## Data Availability

All data are provided in the text.

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
