# Peer review of "Application of Genomic Selection in Beef Cattle Disease Prevention"

_animals, 2025, doi:10.3390/ani15020277_

Round 1
Reviewer 1 Report
Comments and Suggestions for Authors
The authors provide an overview of the application of genomic selection technology to disease prevention, which is expected to provide a new engine for livestock development as technology continues to advance. Before considering publication, I consider the following points of concern:
1. Genomic selection technology improves animal welfare? Which aspect of welfare I consider should be more specific. Because there are many aspects of animal welfare. Here it should be made more concrete.
2. I think the abstract section should be reorganised in terms of ideas, the current version lacks logic. In especially the last sentence about the description of the significance of this study, how is it related to biosafety? The previous section mentions the challenges of cost control, how does that relate to economic benefits? Please reorganise the writing carefully and enhance the logic.
3. The same problem exists with the introduction, where the author's description of each part is completely cut off and illogical, so please do the same to sort out your thoughts and rewrite the introductory part.
4. 4. The authors have read a lot of literature to describe the application of genomic selection technology. However, I believe that the advantages and disadvantages (challenges) of the application of genomic selection technology should be summarised in a table (a diagram would be better). In this way, readers can quickly get the gist of the whole text, which also enhances the readability.
5. Similarly, the author's entire manuscript is devoid of any summary charts and graphs, which I don't think is enough to engage the reader, much less increase the impact of the article.
6. As a Review article, I think the authors have not completely read all the articles in the field, especially some of the latest, high-impact literature seems to be a bit lacking, and I suggest the authors to add it.
Comments on the Quality of English LanguageI have no further comment on this.
Author Response
We thank the reviewer for their constructive comments. We have addressed the comments on a point-by-point basis in the revised version.
___________________________________________________________________________________________________________
The authors provide an overview of the application of genomic selection technology to disease prevention, which is expected to provide a new engine for livestock development as technology continues to advance. Before considering publication, I consider the following points of concern:
- Genomic selection technology improves animal welfare? Which aspect of welfare I consider should be more specific. Because there are many aspects of animal welfare. Here it should be made more concrete.
Authors: Details on animal welfare were expanded
- I think the abstract section should be reorganised in terms of ideas, the current version lacks logic. In especially the last sentence about the description of the significance of this study, how is it related to biosafety? The previous section mentions the challenges of cost control, how does that relate to economic benefits? Please reorganise the writing carefully and enhance the logic.
Authors: The sentences were reorganized as suggested.
- The same problem exists with the introduction, where the author's description of each part is completely cut off and illogical, so please do the same to sort out your thoughts and rewrite the introductory part.
Authors: The introduction is reorganized.
- The authors have read a lot of literature to describe the application of genomic selection technology. However, I believe that the advantages and disadvantages (challenges) of the application of genomic selection technology should be summarised in a table (a diagram would be better). In this way, readers can quickly get the gist of the whole text, which also enhances the readability.
Authors: Diagrams are included
- Similarly, the author's entire manuscript is devoid of any summary charts and graphs, which I don't think is enough to engage the reader, much less increase the impact of the article.
Authors: Figures were included.
- As a Review article, I think the authors have not completely read all the articles in the field, especially some of the latest, high-impact literature seems to be a bit lacking, and I suggest the authors to add it.
Authors: More references were included.
Reviewer 2 Report
Comments and Suggestions for Authors
Application of Genomic Selection in Beef Cattle Disease Pre-2 vention ,I reviewed this paper.the review is very good and important but need some modifications
Abstract lines 36-43 need to be rephrased...
Line 43 there was a typing error please do the correction
Keywords must be reorganized with alphabet order
Introduction is very important and good in typing
Line 49 reference number 1 need more reference in this area
Line 57 more data needed In agriculture production please add more referencene and check references number 4
Line79-89 five compartments authors stated should be reorganized in summarized pattern
Line 91 how can authors identify the degree of disease resistance?
Line 107_113neee more information regarding to food safety please clarify and add this important information
Line 123 how authors could detect the population size?
Line 131-142 what about the references in this paragraph.as this paragraph should be rich of many new references
Line 200!-230 paragraphs related to mastitis and calf affection should be reorganized
Line 236-239 with no references how come?
Lines333-340 there were many typing errors and extra spacing should be avoided
Line 414-428 with only one references please clarify to be clear and should contain many references
Thank you
Author Response
We thank the reviewer for their constructive comments. We have addressed the comments on a point-by-point basis in the revised version.
___________________________________________________________________________________________________________
Application of Genomic Selection in Beef Cattle Disease Prevention ,I reviewed this paper. the review is very good and important but need some modifications.
The authors thank the reviewer for the comments. we have addressed the suggestions provided on a point by point basis.
Abstract lines 36-43 need to be rephrased...
Authors: The sentences were rephrased
Line 43 there was a typing error please do the correction
Authors: None detected however, it was rephrased.
Keywords must be reorganized with alphabet order
Authors: Reorganized
Introduction is very important and good in typing
Line 49 reference number 1 need more reference in this area
Authors; It is a fact; no contracting statements to the fact. So, adding more references won’t change the fact. The reference included is fitting.
Line 57 more data needed In agriculture production please add more referencene and check references number 4
Authors: More references are included.
Line79-89 five compartments authors stated should be reorganized in summarized pattern
Authors: It is rewritten.
Line 91 how can authors identify the degree of disease resistance?
Authors: Explanation Is included.
Line 107_113neee more information regarding to food safety please clarify and add this important information
Authors: The section is expanded.
Line 123 how authors could detect the population size?
Authors: Explanation is included
Line 131-142 what about the references in this paragraph.as this paragraph should be rich of many new references
Authors: More references are included.
Line 200!-230 paragraphs related to mastitis and calf affection should be reorganized
Authors: More information is included
Line 236-239 with no references how come?
Authors: More references are included.
Lines333-340 there were many typing errors and extra spacing should be avoided
Authors: No typing errors or spacing detected; however, the sentences were rephrased.
Line 414-428 with only one references please clarify to be clear and should contain many references
Authors: More references are included.
Round 2
Reviewer 1 Report
Comments and Suggestions for Authors
My comments have been addressed well.